# Modulating Behavioural and Self-Reported Aggression with Non-Invasive Brain Stimulation: A Literature Review

**DOI:** 10.3390/brainsci12020200

**Published:** 2022-01-31

**Authors:** Ruben Knehans, Teresa Schuhmann, David Roef, Hans Nelen, Joost à Campo, Jill Lobbestael

**Affiliations:** 1Department of Criminal Law and Criminology, Faculty of Law, Maastricht University, 6211 LH Maastricht, The Netherlands; david.roef@maastrichtuniversity.nl (D.R.); hans.nelen@maastrichtuniversity.nl (H.N.); j.acampo@maastrichtuniversity.nl (J.à.C.); 2Section Brain Stimulation and Cognition, Department of Cognitive Neuroscience, Faculty of Psychology and Neuroscience, Maastricht University, 6229 ER Maastricht, The Netherlands; t.schuhmann@maastrichtuniversity.nl; 3Department of Clinical Psychological Science, Faculty of Psychology and Neuroscience, Maastricht University, 6229 ER Maastricht, The Netherlands; jill.lobbestael@maastrichtuniversity.nl

**Keywords:** aggression, non-invasive brain stimulation (NIBS), transcranial direct current stimulation (tDCS), continuous theta burst stimulation (cTBS), prefrontal cortex (PFC), ventrolateral prefrontal cortex (VLPFC), dorsolateral prefrontal cortex (DLPFC), approach and withdrawal motivation, Taylor Aggression Paradigm (TAP)

## Abstract

Aggressive behaviour is at the basis of many harms in society, such as violent crime. The efforts to explain, study, and possibly reduce aggression span various disciplines, including neuroscience. The specific brain networks which are involved in the modulation of aggressive behaviour include cortical asymmetry and brain areas such as the dorsolateral prefrontal cortex (DLPFC), the ventrolateral prefrontal cortex (VLPFC), and the ventromedial prefrontal cortex (VMPFC). Recent non-invasive brain stimulation (NIBS) research suggests that both transcranial direct current stimulation (tDCS) and continuous theta burst stimulation (cTBS) can play a role in the modulation of aggressive behaviour by directly changing brain activity. In this review, we systematically explore and discuss 11 experimental studies that aimed to modulate aggressive behaviour or self-reported aggression using NIBS. Out of these 11 studies, nine significantly up- or downregulated aggression by using tDCS or cTBS targeting the DLPFC, VLPFC or VMPFC. The potential applications of these findings span both the clinical and the forensic psychological domains. However, the results are limited by the methodological heterogeneity in the aggression measures used across the studies, and by their generally small sample sizes. Future research should consider improving the localization and specificity of NIBS by employing neuro-navigational instruments and standardized scoring methods.

## 1. Introduction

Aggression is at the root of many societal harms, such as violent crime and bullying, and can have a long-lasting detrimental impact on human life. Understanding the underlying neurobiological mechanisms of aggression facilitates the search for effective ways to reduce aggression. In turn, managing aggression can make society as a whole safer by directly preventing inter-personal harm [1]. Additionally, lowering aggression can reduce medical/therapy costs for individuals, and prevent the costs of legal proceedings [2]. Lastly, treatment-resistant forms of aggression may become treatable via new insights into the underlying mechanisms of aggression. Newer neuroscientific techniques, such as non-invasive brain stimulation (NIBS), are rapidly evolving into instruments with significant clinical relevance for, e.g., treatment-resistant depression [3]. The potential for NIBS to influence aggression is currently unclear because the research is fragmented. The current literature review therefore sheds light on the potential of NIBS to modulate aggression by providing an overview of NIBS research that aimed to influence aggression in experimental research. These findings may fuel discussions on possible (combination) therapies, crime-prevention strategies, or the reduction of recidivism via NIBS.

Aggression refers to behaviour that intends to cause physical or psychological harm to another individual [4]. Aggressive behaviour is dissociable into two motivational types: instrumental/proactive aggression, which is employed in a purposeful instrumental way, and reactive aggression, which takes place spontaneously in reaction to provocation [5,6]. There is a large body of behavioural research that justifies this split in aggression types, such as studies suggesting that a hostile interpretation style and attentional bias toward angry faces is related to reactive aggression, whereas a stronger self-aggression association is linked to proactive aggression [7,8]. Moreover, previous studies have shown that both aggression types display unique correlations with many other behavioural, psychopathological, and etiological concepts (for an overview, see [6,9]). As such, distinguishing reactive from proactive aggression is important.

Unsurprisingly, the origins of aggressive behaviour are highly complex, with contributing factors which are deeply intertwined with the environment, such as upbringing and peer relationships, and also gene-related factors. As a consequence, measuring and affecting aggression poses many challenges, such as the variance in the ways in which aggression is expressed on neural, personal and cultural levels, as well as the ethical concerns regarding research (methodology) on aggression [10]. In spite of this, behavioural assessment methods have been developed during the last few decades that can give reliable and valid proxies of aggressive behaviour while respecting ethical boundaries, such as the Taylor Aggression Paradigm (TAP) [11] and the Voodoo Doll Task [12]. These tasks are often preceded by, for instance, a negative digital interaction between the ostensible ‘opponent’ and the participant, or a cover story involving the opponent. Unbeknownst to the participant, however, the opponent is not a real other person. This negative interaction or information sets the stage for the participant to express aggression toward the opponent during the experiment without causing actual harm to the opponent. In the TAP, the participant competes with a purported opponent in many trials to press a button the fastest. After each trial, the winner can ‘punish’ the loser by administering a sound blast or electric shock, the intensity and length of which is the proxy of aggression [11]. A distinction between reactive- and proactive aggression can also be measured, as the first trial represents unprovoked proactive aggression, whereas subsequent trials represent reactive aggression. In the Voodoo Doll Task, a participant is asked to imagine that a voodoo doll represents a specific person, and is asked to stick pins into the doll. As such, the number of pins stuck in the doll is the proxy for aggression [12].

Neuroimaging studies combined with these aggression experiments fuel (neuro)scientific research on the neural correlates of aggression. On the one hand, aggression is often linked to specific brain regions in the prefrontal cortex (PFC), as many processes related to social and moral behaviour take place there. The PFC includes the dorsolateral prefrontal cortex (DLPFC), the ventrolateral prefrontal cortex (VLPFC), the inferior frontal gyrus (IFG) and the ventromedial prefrontal cortex (VMPFC). The PFC has been shown to be involved in behavioural- and emotional inhibition, as well as the evaluation of threats in contexts of anger and aggression [13]. This is, in part, explained by the ability of the PFC to exert control over regions involved in emotion reactivity, such as the amygdala and insula [13]. Among a large variety of other functions, neuroscientific studies evidence the implication of the DLPFC in antisocial behaviour [14,15], which may in part be explained by the demonstrated involvement of the DLPFC in impaired moral judgment [14]. The VLPFC has been linked to various forms of self-control, including control over immediate temptations, response inhibition, risk-taking behaviour, emotional control, and overall cognitive control [16,17,18,19]. The IFG, in turn, has been shown to be implicated in behaviour inhibition [20]. Lastly, the VMPFC plays a role in anger experience and aggression expressions [13].

On the other hand, another important aggression theory, the cortical asymmetry model, assigns differing broad ‘motivational’ functions to the left and right hemispheres [21]. Specifically, the right hemisphere is thought to be involved in withdrawal motivation and response inhibition, whereas the left hemisphere reflects approach motivation, and is associated with anger and aggression [21]. As such, inducing cortical dominance by specifically upregulating one hemisphere while downregulating the other hemisphere provides a potential target for brain stimulation to affect aggression.

There are various forms of NIBS, among which the most commonly used forms are transcranial electric stimulation (TES), including transcranial direct current stimulation (tDCS), transcranial alternating current stimulation (tACS), and transcranial random noise stimulation (tRNS). Additionally, there is transcranial magnetic stimulation (TMS) and its various sub-forms, such as theta burst stimulation (TBS), including intermittent theta burst stimulation (iTBS) and continuous theta burst stimulation (cTBS). Lastly, there is transcutaneous vagus nerve stimulation (tVNS). The literature indicates that two NIBS instruments are predominantly used in relation to aggression research: tDCS and cTBS. 

tDCS is capable of modulating cortical excitability via electrodes delivering weak anodal or cathodal direct currents over a specific target area on the scalp [22]. Part of this direct current travels through the skull and causes the resting membranes of neurons at the target area to either de- or hyperpolarize. According to the Anodal excitation Cathodal inhibition (AeCi) hypothesis associated with tDCS, anodal tDCS enhances cortical excitability, whereas cathodal tDCS reduces it [23]. However, the coupling of anodal-excitation and cathodal-inhibition effects (AeCi) was mainly shown in motor studies, and might not exist in other stimulation sites [24]. Importantly, these other stimulation sites include the PFC, which is the central region in this literature review [24].

TMS, on the other hand, induces an electrical current in the brain through electromagnetic induction via a TMS coil [25]. A brief surge of current flows through the stimulation coil to produce a changing electric field, which in turn creates an orthogonal changing magnetic field [26]. This magnetic field passes freely through the scalp and skull, and again induces an electric field. When the electric field falls in a conductor, such as brain tissue, current will flow [26]. In this way, TMS can non-invasively induce a current in the brain, which—depending on the frequency of the stimulation—causes neuronal de- or hyperpolarization at the target area. cTBS is a variant of TMS during which short bursts of three pulses at a frequency of around 50 Hz, repeated at 5 Hz intervals, are delivered [27]. This particular stimulation frequency of cTBS generally leads to an inhibition in the targeted area. The stimulation itself only takes 40 s, and the inhibitory effects of this stimulation can last up to 60 min.

In the last few years, a growing body of experimental studies has built on the existing knowledge of neural correlates of aggression, and has used NIBS with the aim to directly modulate (proactive and/or reactive) aggression. The current literature review summarises these studies and gives an overview of the possibility of NIBS to modulate aggression. In total, we found 11 studies that employed either tDCS or cTBS combined with either behaviourally assessed aggression or self-reported aggression.

## 2. Materials and Methods

### 2.1. Literature Review

We followed the PRISMA/Cochrane guidelines for systematic literature searches and the identification of relevant literature. The searches were performed in Worldcat, PubMed and PsycINFO. The keywords were ‘brain stimulation’, ‘aggression’, ‘hostility’, ‘anger’, ‘transcranial direct current stimulation’, ‘transcranial magnetic stimulation’, ‘transcranial alternating current stimulation’, ‘transcranial random noise simulation’, ‘theta burst stimulation’ and ‘transcutaneous vagus nerve stimulation’. The eligibility criteria were experimental NIBS studies in humans that directly measured aggression via either well-established aggression experiments or self-report questionnaires. Only articles published in English were included in the selection process.

The systematic literature search and identification process encompassed the following: 856 studies came up in the databases after the use of the abovementioned search terms. Of these 856 articles, the titles and abstracts were read and judged regarding their relevance in light of the eligibility criteria. After this process, 829 articles were excluded. The remaining 27 articles were read and assessed in detail in order to determine their relevance. Of these articles, 16 were excluded because they focussed on correlates of aggression, as opposed to actual aggression. To be precise, of these 16 articles, ten focused on the recognition and perception of emotions such as anger [28,29,30,31,32,33,34,35,36,37], three articles focused on anger as opposed to behavioural aggression [38,39,40], and the final three articles focused on a correlate of aggression such as impulsivity or cognitive control [41,42,43]. A total of 11 studies fulfilled the criteria to be included in this literature review, as shown in the Prisma flowchart (Figure 1).

### 2.2. Measures of Aggression

In the selected studies, various behavioural aggression paradigms were used: the Taylor Aggression Paradigm (TAP) [11], the Competitive Reaction Time Task (CRTT) [44], the Aggression Infused Ultimatum Game [13], the Voodoo Doll task [12], the Sequence Choosing Task [45], the Tangram Task [45], the Hot Sauce Paradigm [46], and the Social Orientation Paradigm task [27]. As an alternative to behavioural aggression assessment tasks, some studies employed self-report instruments such as the Buss-Perry Aggression Questionnaire [47] and the Reactive Proactive Aggression Questionnaire [6] to measure aggression. Overall, the TAP was the most commonly used aggression task [13,20,27,48,49,50,51].

The TAP is a competitive reaction time game played against an ostensible opponent. The participant is told that the main task goal is to press a button faster than their opponent does, but, in reality, the winner is pre-determined [11]. After each round, the winner determines the intensity and length of a punishment for the loser of that round, which is often a sound blast or an unpleasant sound. The punishment sound set by the participant prior to the first round is typically used as a measure of proactive aggression, while the subsequent punishment sounds set in response to the opponent’s sound blasts index reactive aggression [11]. Another category of behavioural aggression measures is tasks based on the TAP that simulate competition between participants and allow the administration of aversive stimuli [52]. One such variant is the Competitive Reaction Time Task. The TAP is recognised as being internally valid [53], and the CRTT has good convergent validity, discriminant validity and ecological validity [52].

In the Aggression Infused Ultimatum Game, the participant is presented with actual monetary offers from an ostensible opponent [13]. The offers are predetermined to be either fair (40–50% split), medium fair (25–35% split), or unfair (10–20% split). Messages are presented alongside these offers, ranging from non-confrontational messages accompanying fair offers, to intensely provocative messages accompanying unfair offers. After this task, the participants rate their emotions and then perform the TAP. The Aggression Infused Ultimatum Game has sound test–retest reliability and convergent validity [13].

The Voodoo Doll Task measures aggression by asking the participants to imagine that a doll represents a certain person. Subsequently, the participants are invited to stick pins into this doll [12]. The number of pins stuck in the doll is the measure of aggression. The Voodoo Doll Task has excellent reliability, construct validity and convergent validity [12].

The Sequence Choosing Task and the Tangram Puzzle Task require a participant to choose easy or difficult puzzles for a partner to complete. The number of difficult puzzles chosen is the measure of aggression. The Sequence Choosing Task is a validated tool to measure aggression [45]; the Tangram Task is similarly well-validated, and has good convergent and discriminant validity [54].

The participants in the Hot Sauce Paradigm are informed about their opponent’s dislike for spicy foods. In turn, the participant may allocate any amount of hot sauce that the opponent must eat in its entirety. The amount of hot sauce given by the participant in grams is the measure of aggression. The Hot Sauce Paradigm is well-validated in terms of external validity and concurrent validity with self-report questionnaires of aggression [46].

The Social Orientation Paradigm is a monetary game wherein the participant plays with an ostensible partner to gather points [27]. In every round of this game, either an individualistic, cooperative, or aggressive behaviour option can be chosen by the participant, allowing them to receive a point, share half a point, or subtract a point from the partner, respectively. The participant is then provoked by having points subtracted, and the frequency of the participant picking the aggressive option to subtract a point from the partner serves as the measure of aggression. The Social Orientation Paradigm is a well-validated instrument to measure aggression [27].

The Buss-Perry Aggression Questionnaire is scored on a 1 to 5 scale, and measures four dimensions of aggressiveness, namely physical aggression, hostility, anger and verbal aggression [47]. The Buss-Perry Aggression Questionnaire has high internal consistency, test–retest reliability [55], and construct validity [56]. The Reactive Proactive Aggression Questionnaire measures self-reported trait aggression via 12 items gauging proactive aggression and 11 items measuring reactive aggression [6]. High internal validity is shown for reactive aggression, proactive aggression and total aggression [6].

### 2.3. Overview of the Study Methodologies

Table 1 provides an overview of the methodology and results of all of the studies included in the current literature review. Across all 11 studies, N = 602 participants (50% males) were involved, with a range of 18 to 90 participants and an average of 55 participants per study. The percentage of males per study ranged from 21% to 100%. All of the studies included a sham stimulation condition. Out of these studies, seven studies used a between-subject design [14,20,45,48,49,50,57], two used a within-subject design [13,27], and two used mixed-subject designs [22,32]. The NIBS instruments used by the included studies were tDCS (ten studies) [13,14,20,22,45,48,49,50,51,57] and cTBS (one study) [27].

Out of the 11 studies, seven had two conditions: namely, a stimulation condition and sham condition [13,14,22,48,49,51,57]. Four studies had three conditions [20,27,45,50]. In two of these four studies, NIBS on the right hemisphere and NIBS on the left hemisphere were treated as separate experimental conditions [27,45]. In the other two studies, a bilateral setup was used, in which both the anode and cathode were placed frontally, with either the anode placed on the right hemisphere and the cathode on the left hemisphere, or the other way around [20,50].

#### 2.3.1. Stimulation Sites

All ten studies employing tDCS used anodal stimulation on frontal sites, with five studies placing the cathode around the left eyebrow or the supraorbital area [22,45,49,51,57], two studies placing the cathode on the brain region contralateral to the anode as part of a bilateral set-up [20,50], and the three remaining studies placing it either over the occipital cortex Oz, on the posterior base of the neck, or on the right shoulder [13,14,48]. The study employing cTBS, conducted by Perach-Barzilay, stimulated the DLPFC [27].

Five studies stimulated the right DLPFC. Of these, four studies [14,22,49,50] centered the anodal electrode on the international 10–20 EEG system electrode site F4 on the right DLPFC, and one study used cTBS [27]. These four studies employing tDCS placed the cathode either above the left eyebrow, on supraorbital ridges Fp2 and Fp1, on the posterior base of the neck, or on the left DLPFC F3 [14,22,49,50]. Three studies stimulated the left DLPFC with tDCS by applying the anode over F3, with the cathode placed on the posterior base of the neck, the right DLPFC, or supraorbital ridges Fp2 and Fp1 [14,22,50]. Out of these studies, two studies applied anodal stimulation to both the right DLPFC F4 and the left DLPFC F3 simultaneously [14,22].

The right VLPFC was targeted in four studies F6, and the cathodes were placed over the occipital cortex Oz in one study [48] and the contralateral supraorbital area in three studies [45,51,57]. Gallucci et al. not only used anodal tDCS on the right VLPFC F6 but also simultaneously on the left VLPFC by positioning the anode above F5 and the cathode on the contralateral supraorbital area [45]. Only one study applied tDCS to the left (F7) and right IFG (F8) by using anodal stimulation on one hemisphere while using cathodal stimulation on the other, and the other way around [20]. Furthermore, the bilateral VMPFC was only targeted once, with the cathode placed on the right shoulder [13]. The exact stimulation location on the VMPFC resulted from placing the anodal electrode vertically over the forehead, with its side edges equidistant from the eyes, and the lower edge at the nasion line [13]. In the only cTBS study, the right DLPFC was determined by finding the motor spot and then moving the coil 5 cm to the anterior along the mid-sagittal line [27].

#### 2.3.2. Stimulation Lengths and Intensities

The stimulation length varied between 12.5 min and 22 min, with seven studies [13,14,20,27,45,51,57] stimulating for 20 min or longer, two studies [22,50] stimulating for 15 min, and another two studies [48,49] stimulating for 12.5 min. The average stimulation length was 18.1 min. The ramp-up and -down times ranged from 5 s to 30 s, with an average of 14.6 s. The stimulation intensity ranged between 1.5 mA and 2.0 mA, with six studies [13,20,22,45,51,57] employing 1.5 mA and four studies [14,48,49,50] employing 2.0 mA. The electrodes in the tDCS experiments had varying sizes, ranging from 5 × 5 to 5 × 10. No high-density setups were used. The study employing cTBS stimulated at 100% of the active motor threshold with a 70 mm figure-eight coil connected to a Magstim Super Rapid magnetic stimulator with four booster modules, an integrated two-channel EMG amplifier, and system acquisition software [27].

#### 2.3.3. Side Effects

In four studies, no side effects were reported by the participants [27,48,49,50]. In six studies [14,20,22,45,51,57], the reported side effects included itchiness [14], a tingling sensation [14], light-headedness [14], a burning sensation or warmth at the electrode site [14], and a pinching sensation or fatigue [14]. One study reported a minor increase in stress levels in the participants if they sensed a tingling sensation during the stimulation [13]. Out of these side effects, itchiness and a tingling sensation were the most common, and the other side-effects occurred rarely.

## 3. Results

The brain areas targeted by the NIBS studies are the DLPFC [14,22,27,49,50], the VLPFC [45,48,51,57], the IFG [20], and the VMPFC [13]. Out of the 11 included studies, nine (82%) found a significant modulation of behavioural or self-reported aggression as a result of NIBS to either the DLPFC, VLPFC or VMPFC. Of those nine studies, four studies [22,27,49,50] stimulated the DLPFC, four studies [45,48,51,57] stimulated the VLPFC, and one study stimulated the VMPFC [13]. All of these studies used tDCS, except for the study by Perach-Barzilay et al., who employed cTBS [27]. In two studies, namely the studies by Choy et al. and Dambacher et al., the attempted modulation of aggression was non-significant [20,48].

### 3.1. Results for Each Stimulation Site

#### 3.1.1. DLPFC

Five studies targeted the DLPFC to modulate aggression [14,22,27,49,50], of which four (80%) of these studies successfully modulated aggression [22,27,49,50]. Aggression was reduced in two studies [22,49] and increased in two other studies [27,50]. Specifically, the downregulation of aggression was found by Molero-Chamizo et al. [22], who applied anodal tDCS bilaterally over the DLPFC (with anodes on the left and right DLPFC, and cathodes on supraorbital ridges Fp2 and Fp1), which significantly lowered the self-reported aggressiveness scores for physical aggression (*p* = 0.001, *d* = 1.24) and verbal aggression (*p* = 0.002, *d* = 1.11) compared to the sham condition. Dambacher et al. [49] lowered proactive aggression via anodal tDCS on the right DLPFC in males in the experimental condition (*M* = 2.74, *SD* = 1.26, *d* = 1.56) compared to the sham condition (*M* = 4.89, *SD* = 1.50).

The upregulation of aggression was found by Perach-Barzilay et al. [27], who showed a significant increase in aggression in the experimental condition (*M* = 4.688, *SD* = 1.583, *d* = 0.95) where cTBS stimulation was provided on the left DLPFC compared to the condition where participants received right DLPFC stimulation (*M* = 1.563, *SD* = 0.689). The increase in aggression in the left DLPFC group was only marginally significant compared to the sham group (*M* = 3.313, *SD* = 1.324). In the study by Hortensius et al. [50], anodal tDCS of the left DLPFC combined with cathodal tDCS on the right DLPFC increased aggression significantly (F(3, 56) = 6.47, *p* = 0.001, *d* = 1.19) in the TAP when the participants scored higher on insult-related anger, signalling an interaction effect. In this study, both anodal tDCS on the right DLPFC (*p* = 0.47) and the sham condition (*p* = 0.15) led to insignificant results. Lastly, in the study by Choy [14], behavioural aggression in the Voodoo Doll Task was not significantly modulated via anodal tDCS bilaterally on the DLPFC (1.71 = 1.31, *p* = 0.26).

#### 3.1.2. VLPFC

Four studies targeted the VLPFC to modulate aggression; in all four (100%) studies, behavioural aggression was significantly modulated [45,48,51,57]. Aggression was downregulated in three studies [48,51,57] and upregulated in one study [45]. The anodal stimulation of the right VLPFC significantly decreased proactive aggression (*M* = 3.43, *SD* = 1.24, *d* = 1.09) as well as reactive aggression (*M* = 5.19, *SD* = 0.92, *d* = 0.73) compared to the sham condition’s proactive (*M* = 4.82, *SD* = 1.32) and reactive aggression (*M* = 5.96, *SD* = 1.17) [48]. Riva et al. [51] significantly lowered aggression, as measured with the TAP, by applying anodal tDCS over the right VLPFC in the experimental condition (*M* = 5.32, *SD* = 1.96, *d* = 0.42) compared to participants in the sham condition (*M* = 4.63, *SD* = 1.31). Riva et al. [57] also lowered aggression in the Hot Sauce Paradigm via anodal tDCS applied to the right VLPFC, but only in a social-exclusion sub-condition (F(1,76) = 7.28, *p* < 0.009, *d* = 0.62). In contrast, anodal tDCS of the left VLPFC by Gallucci et al. [45] led to a significant increase in behavioural aggression (*M* = 0.24, *SD* = 0.14, *d* = 3.71) compared to the sham condition (*M* = −0.28, *SD* = 0.14).

#### 3.1.3. IFG

Dambacher et al. [20] used anodal tDCS on either the right or left IFG, with the cathode placed on the contralateral IFG, and had the participants perform the TAP. In this study, the behavioural aggression did not significantly differ between the sham condition (*M* = 4.53, *SD* = 1.09) and the experimental condition (*M* = 4.14, *SD* = 1.57).

#### 3.1.4. VMPFC

Gilam et al. [13] significantly reduced behavioural aggression in the TAP by applying anodal stimulation centrally to the VMPFC. Anodal tDCS on the VMPFC lowered the behavioural aggression in the experimental condition (*M* = −1.00, *SD* = 3.38, *d* = 0.95) compared to the sham condition (*M* = 1.92, *SD* = 2.78).

### 3.2. Stimulation Instruments

Ten studies used tDCS [13,14,20,22,45,48,49,50,51,57] and one study used cTBS [27]. Out of the ten studies using tDCS, eight (80%) significantly modulated aggression; more specifically, six downregulated aggression [13,22,48,49,51,57] and two upregulated aggression [45,50], and the effect sizes ranged between *d* = 0.42 and *d* = 3.71 [45,51]. Moreover, four tDCS studies targeted the DLPFC [14,22,49,50], another four targeted the VLPFC [45,48,51,57], one targeted the IFG [20] and one targeted the VMPFC [13]. In the singular study using cTBS targeting the DLPFC, aggression was significantly upregulated (*d* = 0.95) [27].

### 3.3. Cortical Asymmetry

Two studies aimed for cortical asymmetry specifically by upregulating one hemisphere with anodal stimulation while downregulating the other with cathodal stimulation [20,50]. Dambacher [20] aimed to increase the excitability in the left or right IFG with anodal tDCS while decreasing the excitability in the contralateral IFG with cathodal tDCS. Hortensius [50] applied anodal tDCS to either the left or right DLPFC while applying cathodal tDCS to the contralateral DLPFC. One of these two studies successfully modulated aggression (50%) [50]. In the study by Dambacher [20], no significant modulation of aggression occurred. In the study by Hortensius [50], aggression was significantly upregulated by applying anodal tDCS on the left DLPFC while placing the cathode on the right DLPFC, but only in cases where participants also scored high on insult-related anger following insulting feedback (*d* = 1.19). In contrast, studies that primarily modulated participants’ targeted singular brain regions—as opposed to inducing cortical asymmetry—successfully modulated aggression 82% of the time, and had effect sizes ranging from *d* = 0.42 to *d* = 3.71.

### 3.4. Task versus Questionnaire

In ten studies, the aggression outcome was measured at the behavioural level [13,14,20,27,45,48,49,50,51,57]; in one study [22], the aggression outcome was measured using a self-report questionnaire. In eight out of the ten studies that employed NIBS together with a task measuring behavioural aggression, aggression was significantly modulated. More specifically, in five of these eight studies, aggression was significantly downregulated [13,48,49,51,57], and the three others studies showed a significant upregulation of aggression [27,45,50]. In the one study using self-reported aggression as the study outcome, NIBS significantly downregulated aggression [22].

### 3.5. Gender Effects

Some studies also reported brain stimulation by gender effects. For instance, Gallucci [45] found that males generally showed a higher level of aggression (*M* = 0.15, *SD* = 0.11) than females (*M* = −0.18, *SD* = 0.11). Similarly, in Dambacher [49], males were found to show significantly more proactive aggression than females (F(1,62) = 7.142, *p* = 0.010). However, in the study by Gallucci [45], the active stimulation of the left and right DLPFC significantly reduced the difference between the genders; after tDCS, males were no more aggressive than females, as indicated by a significant tDCS-by-gender interaction (F(2,84) = 3.52, *p* = 0.034). Moreover, in Dambacher [49], a significant effect of brain stimulation on proactive aggression was reported, but only in males (*M* = 2.74 and *M* = 4.89, *df* = 11, *p* = 0.018, *d* = 1.55) and not in females (*M* = 3.08 and *M* = 2.66, *df* = 17, *p* = 0.480, *d* = 0.33).

## 4. Discussion

This literature review examined the impact of NIBS on aggression by providing a systematic review of experimental studies that employed NIBS methods prior to having participants engage in a behavioural aggression task, or filling out a self-report aggression questionnaire. We found 11 studies that researched the effect of NIBS on aggressive behaviour. The studies used tDCS and cTBS to inhibit or excite various areas in the PFC.

In neuroscientific studies on aggression, it is assumed that the PFC is the main area involved in aggression regulation [13,14,15,16,17,18,19,20]. Existing research has evidenced the involvement of the PFC, and specific brain regions in the PFC—such as the DLPFC and VLPFC—in behavioural and emotional inhibition, impulsivity, moral judgment, and the evaluation of threats in contexts of anger and aggression [13,14,15,16,17,18,19]. Additionally, the cortical asymmetry model suggested that the direction of the effect of NIBS on aggression depended on hemispheric dominance, with right hemispheric dominance leading to withdrawal motivation and left hemispheric dominance triggering approach motivation [21]. In turn, this fMRI research was the basis for the stimulation sites chosen in the included studies.

Generally, the principle of the studies was to use NIBS on the right PFC, most commonly the right DLPFC [14,22,49,50] or the right VLPFC [45,48,51,57], by exciting this area using anodal tDCS. This was performed using three different procedures. Firstly, one procedure excited the right PFC with anodal tDCS while placing the cathode on a site which was not involved in aggression [13,45,48,49,51,57]. A second procedure applied anodal tDCS to the right PFC and cathodal tDCS to the left PFC, and vice-versa [20,50]. Lastly, there as also the combination of both previous procedures, involving anodal stimulation to the right- and left PFC, with the cathode placed on a site unrelated to aggression [14,22]. The cTBS study inhibited either the right or left DLPFC [27]. Out of the 11 studies applying these NIBS stimulation protocols to areas on the PFC, six studies significantly downregulated aggression [13,22,48,49,51,57] and three studies upregulated aggression [27,45,50].

### 4.1. Contextual Factors

A factor that may influence the (strength of the) effective modulation of aggression is the context/emotional state the participant is in while receiving the brain stimulation. This would be in line with evidence showing that the active functional networks during tDCS influence the effects of the brain stimulation [45]. In this way, the stimulation can differentially impact the results, depending on the context and emotions that the participant is processing during or directly prior to the stimulation. This impact is supported by the study of Hortensius et al. [50], who significantly upregulated aggression, and that of Riva et al. [57], who significantly downregulated aggression via NIBS depending on the contextual factors during and prior to the brain stimulation. In these studies, triggering ‘anger’ or ‘social exclusion’, respectively, prior to NIBS led to the significant up- or downregulation of aggression, whereas aggression was non-significantly modulated in the absence of such contextual factors [50,57]. Relatedly, other contextual factors that may modulate stimulation outcomes are the (experimentally induced) subjective expectations of the participants receiving NIBS [58]. Previous studies indeed showed that positive expectations of NIBS can improve the stimulation outcomes, and negative expectations can hinder the stimulation outcomes [58]. This is important to consider in cases where aggression was not significantly modulated between the sham and stimulation conditions.

These factors may have important implications for clinical and forensic considerations of NIBS. In the absence of a specific context/emotional state, ‘isolated’ NIBS likely is capable of modulating behavioural aggression. Nevertheless, certain emotions, mental states or expectations during or prior to the active brain stimulation could either strengthen or weaken the effects of the stimulation. This signals that context may play a valuable moderating role for the reduction of aggression via NIBS. As such, NIBS protocols aimed at the reduction of aggression should consider presenting effective contextual/emotional factors to the participants, such as inducing highly positive expectations of NIBS and modulating social exclusion or anger [50,57]. In turn, insight into the contextual factors that impact NIBS effectiveness can inspire NIBS treatment protocols in the clinical and forensic domains. For instance, if subjective expectations, social exclusion or anger moderate the effects of NIBS, then it could be valuable for a therapeutic approach to assess these factors of the patient prior to the application of NIBS. As such, the importance of the context of NIBS on its effectiveness may signal an added value of embedding NIBS treatment in a socio-therapeutic context including a therapist, as opposed to a stand-alone NIBS treatment.

### 4.2. Cortical Asymmetry

In line with the cortical asymmetry model, in the study by Hortensius et al. [50] the specific contrast between upregulating the DLPFC in the left hemisphere while downregulating the DLPFC in the right hemisphere was presented as a primary reason for the changes in aggression. Therefore, the cortical asymmetry model seems to explain the modulation of behavioural aggression in that study. However, in a later study by Dambacher et al. [20] that also induced hemispheric dominance, but this time via the IFG instead of the DLPFC, hemispheric dominance failed to lead to the significant modulation of aggression. The authors of that latter study suggest that the hemispheric dominance may not have been successfully induced. An alternative explanation of these null findings may also be that the specific brain region receiving stimulation is vital to the modulation of aggression, as opposed to solely hemispheric dominance. In this case, the stimulation of the DLPFC by Hortensius, as opposed to the IFG by Dambacher, may explain the differential impact both studies found on behavioural aggression. In contrast, the fact that the other studies modulated aggression by stimulating specific brain areas without specific regard for hemispheric dominance suggests that it is possible to modulate behavioural aggression solely by stimulating a specific brain region.

Thus, the successful modulation of aggression has been achieved by two methods: either by inducing hemispheric dominance via NIBS [50], or by stimulating aggression-related brain areas in the PFC [13,22,27,45,48,49,51,57]. However, it remains possible that the modulation of aggression via hemispheric asymmetry in the Hortensius study was also dependent on the specific brain region which was stimulated. This may imply that the cortical asymmetry model may only influence aggression when combined with a brain region which is also known to be able to modulate aggression in isolation, i.e., the DLPFC, VLPFC or VMPFC. In other words; a mix of both models may be most effective. More research is needed to shed light on the relationship between cortical asymmetry, the stimulated brain region, and aggression.

### 4.3. Up- or Downregulation

Six studies showed a downregulation of aggression. In four of these studies (67%), this downregulation was a consequence of targeting the right hemisphere, three times the VLPFC and once the DLPFC, with anodal tDCS [48,49,51,57]; one study (17%) targeted the DLPFC bilaterally with anodal tDCS [22], and another study (17%) targeted the VMPFC bilaterally with anodal tDCS [13]. In contrast, all three studies (100%) that upregulated aggression attained this via NIBS, twice with tDCS [45,50] and once with cTBS [27], on the left hemisphere. These findings align with the cortical asymmetry model and the accompanying motivational directions of the two hemispheres [21]. In line with the notion that withdrawal motivation is associated with the right hemisphere, reductions in aggression were achieved via right-hemispheric stimulation or bilateral stimulation but never solely via left-hemispheric stimulation. Conversely, in line with the association between approach motivation and the left hemisphere, increases in aggression were achieved only via the left hemisphere and not via bilateral or right-hemispheric stimulation. In sum, these results suggest that a decrease in aggression is most likely to occur via the anodal stimulation on the PFC of the right hemisphere, while an increase in aggression is most likely to occur via anodal stimulation on the PFC of the left hemisphere.

### 4.4. Gender Differences

Influences of gender were found in three studies [20,45,49]. Overall, these gender differences were in line with previous literature suggesting that males use more physical aggression whereas females tend to employ more indirect forms of aggression [59,60,61]. One implication of this is that, depending on the behavioural aggression paradigm used and what type of aggression it triggers (direct/physical or indirect/non-physical), certain aggression paradigms may be more prone to the generation of aggression in males as opposed to females, or vice versa. For instance, the TAP—where unpleasant sound blasts index aggression—measures physical aggression, and thus may generate more aggression from males than females [49]. Additionally, Dambacher et al. [49] found that inducing right-hemispheric dominance significantly reduced proactive aggression in males only. Here, the lack of effects of brain stimulation in females may also be explained by floor effects due to females showing lower levels of overt/physical aggression in general [49].

Interestingly, Gallucci et al. [45] found that while males were more generally aggressive than females in the sham condition, this difference disappeared after the left and right NIBS of the VLPFC. One explanation for this is that females handle environmental demands more similarly to males after tDCS, even at the expense of socially acceptable behaviours [45]. Another explanation is that tDCS may be more effective for females than for males, which would be in line with previous studies showing greater effects of tDCS on females’ levels of impulsivity and Theory of Mind [62,63]. However, such differential NIBS effects can only be clearly inferred when the baseline levels of the assessed outcomes are equal in both genders, which was not the case in these studies. Moreover, evidence from previous brain stimulation studies suggests that there is high individual variability in the effects of tDCS on cognition and behaviour [64,65], and gender may further contribute to this inter-variability. Additionally, it is also possible that, e.g., impulsivity and Theory of Mind, and how they are related to aggression, may differ neurologically between genders [65].

### 4.5. Group Differences

Group differences may also play an important role in the effectiveness of NIBS. In the study by Molero-Chamizo et al. [22], self-perceived aggressiveness was reduced in four dimensions following NIBS in murderers but only in three dimensions in non-murderers. In addition, self-reported aggression was lower in all of the dimensions of self-reported aggression in the murderer group after active NIBS stimulation, whereas in the non-murderer group only verbal aggression was lowered. This suggests that factors on a group level (murderers vs non-murderers) may moderate the effects of NIBS on aggression. This differential effect of NIBS may be explained by the unidentical baseline measures of aggressiveness between the groups [22]. Alternatively, it is possible that group-level differences lead to differing responses in the PFC following NIBS [22]. A relevant limitation here is that most of the studies involved healthy participants, often students, which limits the generalizability of these findings to clinical and forensic settings. Future research should shed light on the group-level differences between (predominantly) students and clinical populations with mental disorders, or students compared to forensic populations. This will give insight into whether, and to what degree, group differences may impact the effectiveness of NIBS. Information on these differences is particularly relevant for clinical and forensic treatments.

### 4.6. Consistency of NIBS

The effect sizes in the reported studies varied notably, ranging from d = 0.42 to d = 3.71 [45,51], while other studies failed to significantly modulate aggression via NIBS. Individual differences can contribute to the variability in responses to tDCS [62]. In particular, these may include differences in baseline neuronal state and activity, anatomy, age, and variability in injured brains. The variability in injured brains may also be relevant when considering forensic populations and individuals with various types of psychopathologies. Moreover, personality traits can also modulate the effects of tDCS [66].

In order to achieve more conclusive results on the magnitude of the impact of NIBS on aggression, several improvements are needed. For instance, few studies combine fMRI with NIBS [13]. The lack of functional or structural brain data while employing NIBS hampers the precise localization of specific brain regions for stimulation, which may also impede the effectiveness of NIBS to attain the desired behavioural outcome [13]. In addition, subjective expectations toward NIBS are known to influence the effectiveness of NIBS. Therefore, gauging the participant’s expectations prior to the NIBS treatment and modulating these expectations may improve the effectiveness and consistency of NIBS. Moreover, the lack of fMRI-guidance also limits inferences regarding observed behavioural effects, if behavioural effects occur at all [13]. Neuro-navigator techniques that superimpose a 3D brain model of the participant’s brain over the participant in real-time by the use of a computer monitor or augmented reality goggles may significantly improve the localization of specific brain regions during NIBS stimulation, improving the consistency and possibly the effectiveness of the stimulation. In addition to the localization problems, the specificity of brain stimulation with tDCS is suboptimal, as the resolution of tDCS is limited [48]. Moreover, with tDCS there is a risk that the cathode that is not relevant for the study may indirectly and unintentionally affect and confound the behaviour of the participant [48]. This risk can be reduced by placing the cathode on a brain region which is seemingly uninvolved in aggression, such as ‘Oz’, which is located in the visual cortex. Improvements in the effectiveness of NIBS may also be achieved by combining NIBS methods such as high-resolution tDCS with more focal TMS/cTBS, as the combination of methods has been shown to lead to a high level of anatomical specificity. This anatomical specificity can help future research by revealing and interacting with very specific brain regions and networks, and may uncover the ways in which these interact with aggression. Moreover, such anatomical specificity can lead to the improved consistency and effectiveness of NIBS.

### 4.7. Temporary Effects of NIBS

The short-lived effects of the brain stimulation in the included studies are noteworthy for the clinical and forensic relevance of these findings. As stated, Chen [48] found significant reductions in aggression directly after stimulation between the experimental group and the sham group. However, in a self-report questionnaire 24 h after the experiment (assuming that no residual effects of tDCS remained) there were no significant differences in the aggression scores anymore between the two groups [48]. This may indicate that 24 h is enough for the effect of NIBS to dissipate. A study by Batsikadze et al. [67] suggested that simply increasing the intensity and length of the stimulation does not improve the longevity and effectiveness of tDCS, and might even change the direction of the effects. Instead, repeated stimulation protocols and/or pharmacological interventions such as increased dopamine may effectively prolong the effects of NIBS [68,69]. The ability of NIBS to induce long-term effects in various psychiatric and neurological diseases has been shown many times [70]. For years, TMS has been used to treat illnesses such as major depressive disorder [71]. In order to achieve long lasting effects of NIBS, the regular application of NIBS is necessary. For instance, daily TMS for ten consecutive days produced beneficial effects on depression up to 30 days after the end of the stimulation sessions [72]. The optimal stimulation parameters (the targeting method, stimulation frequency, number of pulses per session, and the number and planning of the sessions) for prolonged effects of NIBS on aggression have not yet been determined.

### 4.8. Limitations

There are several limitations in the existing research on NIBS and aggression. Firstly, the sample sizes in NIBS research on aggression tend to be small, which threatens the power of the statistical analyses. Secondly, due to the limited spatial resolution of tDCS, it is possible that brain areas other than the target area were stimulated, which may confound the findings of the tDCS studies. This limitation is particularly relevant in light of the varying sizes of the electrodes used between the tDCS studies. Thirdly, the TAP/CRTT is the most commonly used behavioural aggression paradigm, but it is criticized for its methodological heterogeneity between studies in relation to the cover stories used, the trial specifications, and the outcome quantification strategies [33]. It is therefore crucial to realise that variations in the methodological setup of the TAP/CRTT can differentially impact the study outcomes. Fourthly, the external validity of the current findings is threatened by the fact that most studies use university students as participants. This may be particularly relevant when considering the clinical or forensic applications of NIBS in patients who are often less educated and younger, and those who suffer from various types of psychopathologies. Nevertheless, the Molero-Chamizo et al. [22] study on prisoners provides evidence for the external validity of NIBS modulating aggression in a forensic population. Fifthly, the AeCi model is a coarse approximation of the principles of causation that govern the brain when positing that anodal stimulation leads to behavioural improvement and cathodal stimulation does the opposite [23]. Particularly in behavioural studies, the final output has a complex and not necessarily direct, linear relationship with this basic mechanism [23]. For instance, in the AeCi model, the cytoarchitecture of the stimulated area is ignored, although anodal stimulation on inhibitory neurons may inhibit a specific cortical network, thus producing cathodal inhibition [23]. Moreover, the AeCi model ignores factors relating to inter-individual variability, such as the functional organization of local circuits, the baseline levels of a given function, genetics, development and aging [62]. In sum, in relation to behaviour as complex as aggression, we must be mindful when considering the AeCi model’s explanation of our findings. Sixthly, the impact of NIBS on aggression may be affected by gender floor effects, as females generally tend to express less aggression. Seventhly, inter-individual differences and contextual factors such as mental states and expectations are also known to affect the effectiveness of NIBS. Eighthly, only three studies used simulation models or fMRI to check the specificity of the NIBS [13,45,51]. Lastly, studies that do not distinguish between proactive and reactive aggression may find null results on total aggression after NIBS, although these distinct forms of aggression may have been significantly modulated in isolation but became insignificant when pooled.

### 4.9. Clinical and Forensic Relevance

Group effects are particularly relevant for the clinical and forensic relevance of NIBS for aggression because the vast majority of NIBS research is performed on students, and the generalizability of such findings to a clinical and forensic population is doubtful. Moreover, the fact that tDCS effects dissipate over time introduces the possibility of naturally restoring the status-quo-ante by ceasing clinical or forensic treatment in case the patient does not respond positively to a NIBS treatment. In addition, aggression research should make use of the latest NIBS developments (such as the use of simulation models like SIMNIBS or fMRI to check stimulation specificity) to improve the consistency and effectiveness of NIBS, and thereby increase its relevance as a clinical and forensic instrument vis-à-vis the existing treatment options. Important considerations here are the side effects of the stimulation; however, based on the current research, these side effects seem to be minimal. Lastly, this literature review has shown that the stimulation of the right PFC with an excitatory tDCS protocol consistently downregulates aggression, and the stimulation of the left PFC with an excitatory tDCS protocol or cTBS upregulates aggression. Although the direction of the effects is consistently linked to specific hemispheres, and though this is vital for clinical considerations, these insights are based on a limited number of studies.

### 4.10. Future Research

Future research should further explore the relationship between the cortical asymmetry model, the areas in the PFC involved in aggression, and behavioural aggression. Moreover, future studies that employ the TAP/CRTT should consider the use of an empirically derived scoring method [33] that averages all of the pre-provocation trials against all of the post-provocation trials. In order to improve the homogeneity across TAP/CRTT studies further, future studies should be aware of and critically reflect on cover story aspects and trial specifications, as these can also impact the results of the TAP when, for instance, participants question the realness of their opponent. Moreover, more studies should be performed on larger samples and heterogenous populations, including different cultures, genders, and clinical and forensic populations. An interesting avenue for research that is also relevant for clinical applications of NIBS is the exploration of whether tools such as virtual reality can be employed to provide or strengthen the contextual factors in order to positively enhance the effects of NIBS on participants. Lastly, the papers reviewed for this paper only included those directly targeting aggression, not those targeting aggression correlates. There has been an increase of the latter study types over the past few years. For the most part, these studies suggest that NIBS can also modulate such correlates of aggression. For instance, NIBS can be used to reduce attention toward or modulate interpretations of angry faces [28,29,30,31,32,33,34,35,36,37]. Relatedly, NIBS has also successfully modulated impulsivity [41], and has led to reduced feelings of anger in individuals with e.g., borderline personality disorder or depressive disorder [42]. These latter studies focused predominantly on areas in the PFC such as the medial-frontal cortex and the IFG. As such, it might be fruitful to modulate these brain regions related to correlates of aggression, as they may also indirectly moderate behavioural aggression.

## 5. Conclusions

In conclusion, the current literature review demonstrates that aggression can be successfully up- and downregulated using NIBS. While many aspects of the research on the relationship between NIBS and aggression still need to be improved upon, this research reveals an exciting starting point, and makes suggestions for further research on the modulation of aggression via NIBS.

## Figures and Tables

**Figure 1 brainsci-12-00200-f001:**
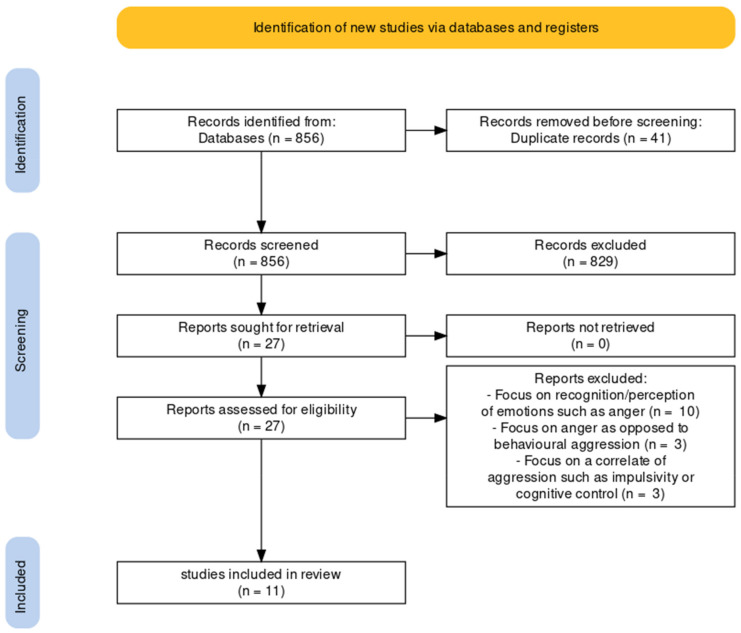
Prisma flowchart showing the selection process of the articles used in the review. Note: The terms ‘report’, ‘article’ and ‘study’ are used interchangeably.

**Table 1 brainsci-12-00200-t001:** Overview for each study of the methodology and results.

Authors and Publication Year	Participants and Gender Division	Design	NIBS Method	Stimulated Brain Region	NIBS Location	Stimulation Type	Stimulation Length	Stimulation Intensity	Side Effects	Aggression Measures	Sham Condition Total Aggression	Sham Condition Reactive Aggression (*M*/*SD*)	Sham Condition Proactive Aggression (*M*/*SD*)	Experimental Condition Total Aggression (*M*/*SD*)	Experimental Condition Reactive Aggression (*M*/*SD*)	Experimental Condition Proactive Aggression (*M*/*SD*)	Cohen *d*
Chen (2019) [48]	32 (50% males)	Sham-controlled, RCT, double-blind, between-subject design	tDCS	Right VLPFC	Anode over right VLPFC (F6), cathode over occipital cortex (Oz)	Anodal or sham	12.5 min (20 s ramp up/down)	2.0 mA	None reported.	TAP	N/A	*M* = 5.96 + −1.17	*M* = 4.82 + −1.32	N/A	*M* = 5.19 + −0.92 *	*M* = 3.43 + −1.24 *	Reactive aggression *d* = 0.73 (−) Proactive aggression *d* = 1.09 (−)
Choy et al. (2018) [14]	81 (44% males)	Sham-controlled, RCT, double-blind, between-subject design	tDCS	Bilateral DLPFC	Anodes bilaterally over left DLPFC (F3) and right DLPFC (F4), cathode: posterior base of the neck	Anodal or sham	20 min (30 s ramp up and 2 s ramp down)	2.0 mA	Mild physical side effects reported.	Voodoo Doll Task	N/A	N/A	N/A	F(1,71) = 1.31, *p* = 0.26	N/A	N/A	-
Dambacher et al. (2015) [49]	32 (63% males)	Sham-controlled, RCT, between-subject design	tDCS	Right DLPFC	Anode: right DLPFC (F4), cathode: left eyebrow	Anodal or sham	12.5 min (20 s ramp up/down)	2.0 mA	None reported.	TAP	4.00 + −1.33	*M* = 4.13 + −1.48	3.49 + −1.72	3.84 + −1.16	*M* = 4.07 + −1.25	*M* = 2.93 + −1.25 **	Proactive aggression *d* = 1.56 ** (−)
Dambacher et al. (2015) [20]	64 (61% males)	Sham-controlled, RCT, between-subject design, two experimental conditions	tDCS	Right or left IFG	Anode: right IFG (F8), cathode: left IFG (F7).	Anodal and cathodal or sham	21.75 min (20 s ramp up/down)	1.5 mA	Mild physical side effects reported.	TAP	4.53 + −1.09	4.83 + −1.24	3.36 + −1.43	4.14 + −1.57	4.4 + −1.65	3.10 + −1.69	-
Gallucci et al. (2020) [45]	90 (50% males)	Sham-controlled, RCT, double-blind, between-subject design, two experimental conditions	tDCS	Right or left VLPFC	Anode: right VLPFC (F6) or anode: left VLPFC (F5); sham, randomized target area (left/right); cathodal reference electrode placed over contralateral supraorbital area	Anodal or sham	20 min (10 s ramp up/down)	1.5 mA	Mild physical side effects reported.	CRTT, Sequence choosing task, Tangram Task	*M* = −0.28 + −0.14	N/A	N/A	*M* = 0.24 + −0.14 *	N/A	N/A	Total aggression *d* = 3.71 (+)
Gilam et al. (2018) [13]	25 (40% males)	Sham-controlled, RCT, within-subject design	tDCS	VMPFC	Anode: placed vertically over forehead (side-edges equidistant from eyes, lower edge at nasion line, fixed with head sweat-band), cathode: extra-cephalically placed on right shoulder and fixed with elastic band-aid	Anodal or sham	22 min (30 s ramp up/down)	1.5 mA	Minor increase in stress level in participants who sensed the stimulation.	TAP	*M* = 1.92 + −2.78	N/A	N/A	M = −1.00 + −3.38 *	N/A	N/A	Total aggression *d* = 0.95 (−)
Hortensius et al. (2012) [50]	60 (50% males)	Sham-controlled, RCT, double-blind, between-subject design, two experimental conditions	tDCS	Right or left DLPFC	Bilateral montage: anode left DLPFC (F3) and cathode right DLPFC (F4) and vice-versa	Anodal and cathodal or sham	15 min (5 s ramp up/down)	2 mA	None reported.	TAP	N/A	N/A	N/A	F(3, 56) = 6.47, *p* = 0.001 ***	N/A	N/A	Total aggression *d* = 1.19 *** (+)
Molero-Chamizo et al. (2019) [22]	41 (100% males)	Sham-controlled, RCT, mixed design (within- and between-subject)	tDCS	Bilateral DLPFC	Anodes: bilaterally over left DLPFC (F3) and right DLPFC (F4), cathodes: supraorbital ridges (Fp2 and Fp1)	Anodal or sham	15 min (10 s ramp up/down)	1.5 mA	Mild physical side effects reported.	Buss Perry Aggression Questionnaire	N/A	N/A	N/A	*p* = 0.001 physical aggression * *p* = 0.002 verbal aggression *	N/A	N/A	Physical aggression *d* = 1.24 (−) Verbal aggression *d* = 1.11 (−)
Perach-Barzilay et al. (2013) [27]	18 (77% males)	Sham-controlled, RCT, within-subject design, two experimental conditions	cTBS	Right or left DLPFC	5 cm rule first identifying the motor spot (site in which a single TMS produces maximal amplitude of motor response of the APB muscle) and then moving the coil 5 cm to the anterior along mid-sagittal line	Inhibitory cTBS	20 min	triple-pulse 50 Hz bursts delivered at a rate of 5 Hz (200 ms between bursts)	None reported.	Social orientation Paradigm	*M* = 3.31 + −1.32	N/A	N/A	M = 4.69 + −1.58 *	N/A	N/A	Total aggression *d* = 0.95 (+)
Riva P, et al. (2015) [57]	80 (21% males)	Sham-controlled, RCT, between-subject design	tDCS	Right VLPFC	Anode: right VLPFC (F6), cathode: contralateral supraorbital area	Anodal or sham	20 min (8 s ramp up and 5 s ramp down)	1.5 mA	Mild physical side effects reported.	Hot Sauce Paradigm	N/A	N/A	N/A	F(1,76) = 2.30, *p* < 0.14 ****	N/A	N/A	Total aggression *d* = 0.62 **** (−)
Riva P, et al. (2017) [51]	79 (53% males)	Sham-controlled, RCT, mixed design (sham-controlled between-subject, aggression type within-subject)	tDCS	Right VLPFC	Anode: right VLPFC (F6), cathode: contralateral supraorbital area	Anodal or sham	20 min (8 s ramp up/down)	1.5 mA	Mild physical side effects reported.	TAP	*M* = 4.63 + −1.31	N/A	N/A	M = 5.32 + −1.96 *	N/A	N/A	Total aggression *d* = 0.42 (−)

^1^ Note: The N/A and F-values are included in the table due to the absence of descriptive data in the respective articles. Mild physical side effects: Some cases of itchiness, discomfort at the electrode sites, or a tingling sensation. RCT = randomized controlled trial; tDCS = transcranial direct current stimulation; cTBS = continuous theta burst stimulation; VLPFC = ventrolateral prefrontal cortex; DLPFC = dorsolateral prefrontal cortex; VMPFC = ventromedial prefrontal cortex; IFG = inferior frontal gyrus; mA = milliamp; TAP = Taylor Aggression Paradigm; CRTT = Competitive Reaction Time Task. * = Significant at the *p* < 0.05 level. ** = A significant result only for males; *M* = 2.74 + −1.26, Cohen *d* = 1.56. *** = A significant result only when the participants scored high on insult-related anger. **** = A significant result only of NIBS reducing aggression in a social exclusion sub-condition: F(1,76) = 7.28, *p* < 0.009, *d* = 0.6. (−) Aggression was downregulated; (+) aggression was upregulated.

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
