# Peer review of "Modulating Behavioural and Self-Reported Aggression with Non-Invasive Brain Stimulation: A Literature Review"

_brainsci, 2022, doi:10.3390/brainsci12020200_

Round 1

Reviewer 1 Report

The review collects evidence on the effects of Non-Invasive Brain Stimulation (NIBS) on aggression both considering behavioral outcomes and self-reported measures. Out of the eleven studies the review explored, nine significantly modulated aggression with tDCS or cTBS over DLPFC, VLPFC or VMPFC, while two failed in modulating aggression. The Authors conclude that NIBS over these cortical areas could be used to up-regulate or down-regulate aggressivity with important implications for clinical and forensic domains.  The aim of this contribution is original and interesting. The manuscript is well-written and structured, surely relevant to the field. However, few ameliorations are needed, and some comments should be addressed. Since numeration of pages above are incorrect, in this report I will refer to the correct page numbers.

Some specific comments:

  • Citations in Reference are incorrect. Two “citation 1” are present at the beginning of the References, with consequent wrong assignation of numbers to all the citations. Please carefully control citations and their numbers. Citation 30  lacks a “b” in the year
  • Reformulate this phrase which to me is unclear: “Out of six studies that downregulated aggression, four times this resulted anodal right hemispheric NIBS (67%) [29,30,32,38], once via bilateral NIBS (17%) [22] and once via bilateral VMPFC stimulation (17%) [13].” (page 15, “Up- and dowregulation”, line 1).

Some more general comments:

  • The description of NIBS, in some cases, is superficial and imprecise. E.g., the statements “Anodal stimulation increases brain activation at the target area while cathodal stimulation decreases it” (page 3, line 47) and “These results suggest that the right hemisphere should be targeted with anodal NIBS to reduce aggression and the left hemisphere should be targeted with anodal NIBS to increase aggression.” (page 15, “Up- or downregulation”, last three lines) are only marginally true, even more if we consider that the cortical sites discussed in this review are mainly prefrontal cortex areas. Here Authors are using the term "NIBS", but maybe they refer to tDCS anodal and cathodal stimulation. However, it was shown that the Ae-Ci model (that is, an enhancement in activation due to anodal stimulation and a worsening due to cathodal stimulation) is reliable when stimulating motor cortex, while it is imprecise when targeting other prefrontal sites. Authors should discuss this topic with caution and present more precisely the possible realistic effects of anodal and cathodal stimulation when targeting PFC.
  • Also, TMS is superficially introduced at page 3. Authors should better explain basic neurobiological mechanisms of both TMS and tDCS to allow the reader better understanding the following section devoted to the modulation of aggression.
  • I found a little confounding the list of tasks in the Method section soon after the description of the literature search criteria (pages 5-6); I would insert all the tasks described in a separate paragraph. Also, the repetition of the montage settings in three sections (“Overview of study methodology”, “Stimulation sites”, “Results”) is confounding: Authors should try to simplify where possible, explaining the montage in a separated paragraph and then reporting only the results of each study.
  • It is not clear why out of 856 articles, only few met the criteria for being included in the review. In particular, it is unclear the expression “focused on correlates of aggression as opposed to actual aggression” (page 4, “Literature review”, line 19). I suggest Authors to better explain what they mean.

Finally, a general concept comment:

  • Authors at the end report some factors which could undermine the effects of the stimulation on aggression. As other domains in which NIBS are administered, and particularly when stimulating PFC, it is important to mention also the role of individual differences and subjective expectations. It is well known that properties of the stimulated brain tissue might affect the effectiveness of the stimulation. Moreover, in some cases, experimentally induced expectations can interact with tDCS . It is also possible that tDCS might be more effective in participants reporting certain personality traits  or displaying positive expectations about the stimulation. This might also explain the lack of difference between real stimulation and sham in some studies mentioned by the Authors (page 11, page 12). These aspects should be highlighted in “Contextual Factors” (page 14), “Consistency of NIBS” (page 17) and “Limitations” (page 18).

Author Response

Dear editor,

We are grateful for your invitation to revise and resubmit our manuscript ‘Modulating behavioural- and self-reported aggression with non-invasive brain stimulation: A literature review’, brainsci-1547015. Below, we provide a detailed point-by-point reply to all the issues that were raised by the two reviewers.

Please note that, as pointed out by reviewer #1, in the original manuscript the page numbers and line numbers did not properly show. They have now been corrected in the revised manuscript and so now the correct page numbers and line numbers are referred to when addressing the comments of the reviewers to ease verification in the revised manuscript. Please find attached the manuscript with all changes highlighted.

Reply to reviewer #1:

The review collects evidence on the effects of Non-Invasive Brain Stimulation (NIBS) on aggression both considering behavioral outcomes and self-reported measures. Out of the eleven studies the review explored, nine significantly modulated aggression with tDCS or cTBS over DLPFC, VLPFC or VMPFC, while two failed in modulating aggression. The Authors conclude that NIBS over these cortical areas could be used to up-regulate or down-regulate aggressivity with important implications for clinical and forensic domains.  The aim of this contribution is original and interesting. The manuscript is well-written and structured, surely relevant to the field. However, few ameliorations are needed, and some comments should be addressed. Since numeration of pages above are incorrect, in this report I will refer to the correct page numbers.

Reply: We appreciate that the reviewer finds our contribution well-written, well-structured, and interesting and relevant to the field. The ameliorations that the reviewer suggested have been taken to heart and have been carefully implemented throughout the manuscript. Please see our point-by-point replies below.

Some specific comments:

  • Citations in Reference are incorrect. Two “citation 1” are present at the beginning of the References, with consequent wrong assignation of numbers to all the citations. Please carefully control citations and their numbers. Citation 30 lacks a “b” in the year

Reply: We thank the reviewer for pointing this out. All citations have been adapted and checked.

  • Reformulate this phrase which to me is unclear: “Out of six studies that downregulated aggression, four times this resulted anodal right hemispheric NIBS (67%) [29,30,32,38], once via bilateral NIBS (17%) [22] and once via bilateral VMPFC stimulation (17%) [13].” (Page 15, “Up- and dowregulation”, line 1).

Reply: We thank the reviewer for pointing this out. We have rephrased this sentence to more clearly put forward the relationship between downregulation of aggression and NIBS stimulation location:

Reformulation line 618: ‘Six studies showed a downregulation of aggression. In four of these studies (67%), this downregulation was a consequence of targeting the right hemisphere, three times the VLPFC and once the DLPFC, with anodal tDCS [48,49,51,57], one study (17%) targeted the DLPFC bilaterally with anodal tDCS [22], and another study (17%) targeted the VMPFC bilaterally with anodal tDCS [13].’

Some more general comments:

  • The description of NIBS, in some cases, is superficial and imprecise. E.g., the statements “Anodal stimulation increases brain activation at the target area while cathodal stimulation decreases it” (page 3, line 47) and “These results suggest that the right hemisphere should be targeted with anodal NIBS to reduce aggression and the left hemisphere should be targeted with anodal NIBS to increase aggression.” (page 15, “Up- or downregulation”, last three lines) are only marginally true, even more if we consider that the cortical sites discussed in this review are mainly prefrontal cortex areas. Here Authors are using the term "NIBS", but maybe they refer to tDCS anodal and cathodal stimulation. However, it was shown that the Ae-Ci model (that is, an enhancement in activation due to anodal stimulation and a worsening due to cathodal stimulation) is reliable when stimulating motor cortex, while it is imprecise when targeting other prefrontal sites. Authors should discuss this topic with caution and present more precisely the possible realistic effects of anodal and cathodal stimulation when targeting PFC.

Reply: We are grateful that the reviewer sheds light on this imprecision. Accordingly in the revised manuscript we are more tentative and accurate when describing NIBS and additionally incorporate a discussion of the AeCi model to add depth to the reader’s understanding of NIBS. To accommodate this improvement, we have made changes at various parts throughout the manuscript.

AeCi model introduced in line 147: ‘According to the Anodal excitation Cathodal inhibition (AeCi) hypothesis associated with tDCS, anodal tDCS enhances cortical excitability whereas cathodal tDCS reduces it [23]. However, the coupling of anodal-excitation and cathodal-inhibition effects (AeCi) was mainly shown in motor studies and might not exist in other stimulation sites [24]. Importantly, these other stimulation sites include the PFC, which is the central region in this literature review [24].’

Moreover, in the “Limitations” section the AeCi model is discussed with a focus on cytoarchitecture, interpersonal variability and the limited applicability of the AeCi model to behavioural studies. This last point is of course important in relation to our literature review on behavioural aggression, and so it is rightly emphasized as such.

AeCi model shortcomings in the limitations section in line 783: ‘Fifthly, the AeCi model is a coarse approximation of the principles of causation that govern the brain when positing that anodal stimulation leads to behavioural improvement and cathodal stimulation does the opposite [23]. Particularly in behavioural studies, the final output has a complex and not necessarily direct linear relationship with this basic mechanism [23,73,74]. For instance, in the AeCi model the cytoarchitecture of the stimulated area is ignored although anodal stimulation on inhibitory neurons may inhibit a specific cortical network, thus producing cathodal inhibition [23]. Moreover, the AeCi model ignores factors relating to interindividual variability such as functional organization of local circuits, baseline levels of a given function, genetics, development and aging [62]. In sum, in relation to behaviour as complex as aggression, we must be mindful when considering the AeCi model explanation of our findings.’

  • Also, TMS is superficially introduced at page 3. Authors should better explain basic neurobiological mechanisms of both TMS and tDCS to allow the reader better understanding the following section devoted to the modulation of aggression.

Reply: We thank the reviewer for this suggestion. Indeed, more details on the neurobiological mechanisms of TMS and tDCS allow the reader to better grasp the possibilities and limitations of NIBS, particularly in relation to aggression. We have added details on the neurobiological mechanisms of both TMS and tDCS.

First, the TMS description has been complemented by a section describing the neurobiological mechanism of TMS.

Neurobiological mechanism of TMS added in line 156: ‘A brief surge of current flows through the stimulation coil to produce a changing electric field, which in turn creates an orthogonal changing magnetic field [26]. This magnetic field passes freely through the scalp and skull and again induces an electric field. When the electric field falls in a conductor, such as brain tissue, current will flow [26]. Hereby TMS can non-invasively induce a current in the brain, which, depending on the frequency of the stimulation, causes neuronal de- or hyperpolarization at the target area.’

Second, to further add to the reader’s understanding of the NIBS methods presented in this manuscript, a short description of the direction of effects of the cTBS frequency that is used by the one cTBS study included in the manuscript has also been added.

Short explanation of the cTBS frequency added in line 166: ‘This particular stimulation frequency of cTBS generally leads to an inhibition in the targeted area.’

  • I found a little confounding the list of tasks in the Method section soon after the description of the literature search criteria (pages 5-6); I would insert all the tasks described in a separate paragraph. Also, the repetition of the montage settings in three sections (“Overview of study methodology”, “Stimulation sites”, “Results”) is confounding: Authors should try to simplify where possible, explaining the montage in a separated paragraph and then reporting only the results of each study.

Reply: We thank the reviewer for pointing this out. We gladly incorporate the reviewer’s suggestions, as we think it indeed contributes to a more clear and concise presentation of our methodology.

Firstly, following the reviewer’s suggestion, we have added a separate header at line 209 called ‘2.2. Measures of Aggression’ after the description of the literature search criteria. Under this new header, all the aggression tasks are now grouped and described.

Secondly, we appreciate and acknowledge the reviewer’s remark in relation to the repetition of the montage settings. Indeed, discussing the montage settings at three occasions was superfluous and so the section about montage settings under ‘Overview of study methodologies’ has now been moved and incorporated in the section ‘Stimulation sites’ below it. This section now organically starts by giving a broader overview of anode and cathode locations in all studies and the paragraphs that follow go into more detail about the anode/cathode location per brain region that is stimulated in the included studies. In its revised form, the manuscript brings up and discusses montage settings at two occasions: in section ‘Stimulation sites’ and section ‘Results’. We believe that having these two sections concern montage settings is justifiable as these sections discuss the montage settings in a different light, once methodologically and once in relation to the findings of the literature review.

Moved the following section from ‘Overview of study methodologies’ to ‘Stimulation sites’ line 306: ‘All ten studies employing tDCS used anodal stimulation on frontal sites with five studies placing the cathode around the left eyebrow or the supraorbital area [22,45,49,51,57], two studies placing the cathode on the brain region contralateral to the anode as part of a bilateral set-up [20,50] and the three remaining studies placed it either over the occipital cortex Oz, the posterior base of the neck or on the right shoulder [13,14,48]. The study employing cTBS by Perach-Barzilay stimulated the DLPFC [27].’

After moving this text out of its original paragraph, the phrase: ‘The NIBS instruments used by the included studies are tDCS (ten studies) [13,14,20,22,45,48-51,57] and cTBS (one study) [27].’ was the only one left of the paragraph and so has been moved to the paragraph above it in line 294.

  • It is not clear why out of 856 articles, only few met the criteria for being included in the review. In particular, it is unclear the expression “focused on correlates of aggression as opposed to actual aggression” (page 4, “Literature review”, line 19). I suggest Authors to better explain what they mean.

Reply: We thank the reviewer for this comment. We agree that it is valuable to explain in more detail why 16 articles were excluded from the selection after making it through the first ‘filter’. After another good look at these articles, we have identified three categories that reflect the reasons of those articles that did not make it into the final 11. These three categories are added in the text in line 199. In addition, these three categories have replaced the text “Study focuses on a correlate of aggression (N=16)” in the PRISMA flowchart (Figure 1). The three categories are:

*focus on recognition/perception of emotions such as anger – 10 articles

*focus on anger as opposed to behavioural aggression – 3 articles.

*focus on a correlate of aggression such as impulsivity or cognitive control – 3 articles.

The three categories are added to the text in line 199: ‘To be precise, of these 16 articles, ten focused on the recognition and perception of emotions such as anger [28-37], three articles focused on anger as opposed to behavioural aggression [38-40], and the final three articles focused on a correlate of aggression such as impulsivity or cognitive control [41-43].’

Finally, a general concept comment:

  • Authors at the end report some factors which could undermine the effects of the stimulation on aggression. As other domains in which NIBS are administered, and particularly when stimulating PFC, it is important to mention also the role of individual differences and subjective expectations. It is well known that properties of the stimulated brain tissue might affect the effectiveness of the stimulation. Moreover, in some cases, experimentally induced expectations can interact with tDCS. It is also possible that tDCS might be more effective in participants reporting certain personality traits or displaying positive expectations about the stimulation. This might also explain the lack of difference between real stimulation and sham in some studies mentioned by the Authors (page 11, page 12). These aspects should be highlighted in “Contextual Factors” (page 14), “Consistency of NIBS” (page 17) and “Limitations” (page 18)

Reply: We thank the reviewer for pointing out these relevant limitations, which indeed deserve attention in our manuscript. Therefore, they have been incorporated in the manuscript in the sections that the reviewer highlights.

Firstly, the aspect ‘subjective expectations’ has been added in the section ‘Contextual Factors’. When discussing this aspect, a link is also made with the lack of difference between sham and stimulation conditions in some studies.

Added subjective expectations in ‘Contextual factors’ line 557: ‘Relatedly, other contextual factors that may modulate stimulation outcomes are the (experimentally induced) subjective expectations of participants receiving NIBS [58]. Previous studies indeed show that positive expectations of NIBS can improve stimulation outcomes and negative expectations can hinder stimulation outcomes [58]. This is important to consider in cases where aggression was not significantly modulated between sham and stimulation conditions.’

Small changes in ‘Contextual factors’ accommodating the above addition:

In line 574 ‘expectations’ have also been included: ‘As such, NIBS protocols aimed at reducing aggression should consider presenting effective contextual/emotional factors to the participants, such as social exclusion or anger’ reformulated into: ‘As such, NIBS protocols aimed at reducing aggression should consider presenting effective contextual/emotional factors to the participants, such as inducing high positive expectations of NIBS and modulating social exclusion or anger’.

In line 578 ‘expectations’ are now also included: ‘For instance, if social exclusion or anger moderate effects of NIBS then it can be valuable for a therapeutic approach to assess these factors of the patient prior to applying NIBS.’ reformulated into: ‘For instance, if subjective expectations, social exclusion or anger moderate effects of NIBS then it can be valuable for a therapeutic approach to assess these factors of the patient prior to applying NIBS’.

Secondly, ‘subjective expectations’ is added to the section ‘Consistency of NIBS’. In addition, we added a suggestion to try control for these factors to increase NIBS effectiveness and consistency, in line with the topic of this particular section.

Added subjective expectations in ‘Consistency of NIBS’ in line 711: ‘In addition, subjective expectations toward NIBS are known to influence NIBS effectiveness. Therefore, gauging the participant’s expectations prior to NIBS treatment and modulating these expectations may improve NIBS effectiveness and consistency.’

Thirdly, inter-individual variability already came up in the section “Gender differences” Line 667: “Moreover, evidence from previous brain stimulation studies suggest there is high individual variability in tDCS effects on cognition and behaviour [64,65] and gender may further contribute to this inter-variability”. However, individual variability is now made more explicit and prominent as a limitation by presenting individual variability in both the ‘Consistency of NIBS’ and the ‘Limitations’ section.

Added influence of individual differences and personality traits on NIBS in ‘Consistency of NIBS’ in line 698: ‘Individual differences can contribute to the variability in responses to tDCS [62]. Particularly differences in baseline neuronal state and activity, anatomy, age and variability in injured brains. The variability in injured brains may be also relevant when considering forensic populations and individuals with various types of psychopathologies. Moreover, personality traits can also modulate the effects of tDCS [66].’

Added inter-individual differences and contextual factors such as expectations to the limitations section in line 799: ‘Seventhly, inter-individual differences and contextual factors such as expectations and mental states are also known to affect the effectiveness of NIBS.’

We hope that we were able to address the feedback of the reviewers in a satisfactory way. We look forward to hearing your opinion on the suitability of our revised manuscript in your special issue on Dimensions of Pathological Aggression: From Neurobiology to Therapy, for Brain Sciences.

Sincerely, also on behalf of the co-authors

Ruben Knehans

References:

23. Fertonani A., Miniussi C. (2017). Transcranial Electrical Stimulation: What We Know and Do Not Know About Mechanisms. The Neuroscientist. 23(2), 109-123.

24. Jacobson L., Koslowsky M., Lavidor M. (2012). tDCS polarity effects in motor and cognitive domains: a meta-analytical review. Experimental Brain Research. 216, 1-10.

26. Post A., Keck M.E. (2001). Transcranial magnetic stimulation as a therapeutic tool in psychiatry: what do we know about the neurobiological mechanisms? Journal of Psychiatric Research. 35, 193-215.

28. Vonck S., Patrick Swinnen S.P., Wenderoth N., Alaerts K. (2015). Effects of transcranial direct current stimulation on the recognition of bodily emotions from point-light displays. Frontiers in Human Neuroscience. 9:438, 1-8.

29. Janik A.B., Rezlescu C., Banissy M.J. (2015). Enhancing Anger Perception With Transcranial Alternating Current Stimulation Induced Gamma Oscillations. Brain Stimulation. 8(6), 1138-1143.

30. Yang T., Banissy M.J. (2017). Enhancing anger perception in older adults by stimulating inferior frontal cortex with high frequency transcranial random noise stimulation. Neuropsychologia. 102, 163-169.

31. Ferrucci R., Giannicola G., Rosa M., Fumagalli M., Boggio P.S., Hallett M., Zago S., Priori A. (2012). Cerebellum and processing of negative facial emotions: cerebellar transcranial DC stimulation specifically enhances the emotional recognition of facial anger and sadness. Cognition and Emotion. 5, 786-799.

32. van Honk, J., Hermans, E. J., d’alfonso, A. A., Schutter, D. J., van Doornen, L., and de Haan, E. H. (2002a). A left-prefrontal lateralized, sympathetic mechanism directs attention towards social threat in humans: evidence from repetitive transcranial magnetic stimulation. Neuroscience Letters. 319, 99–102.

33. d’Alfonso A.A., Van Honk J., Hermans E., Postma A., de Haan E.H. (2000). Laterality effects in selective attention to threat after repetitive transcranial magnetic stimulation at the prefrontal cortex in female subjects. Neuroscience Letters. 280(3), 195-198.

34. Donhauser P.W., Belin P., Grosbras M. (2014). Biasing the perception of ambiguous vocal affect: a TMS study on frontal asymmetry, Social Cognitive and Affective Neuroscience. 9(7), 1046–1051.

35. Harmer C.J., Thilo K.V., Rothwell C.J., Goodwin G.M. (2001). Transcranial magnetic stimulation of medial-frontal cortex impairs the processing of angry facial expressions. Nature Neuroscience. 4(1), 17-18.

36. Schutter D.J.L.G., Van Honk J., Laman M., Vergouwen A.C., Koerselman F. (2010). Increased sensitivity for angry faces in depressive disorder following 2 weeks of 2-Hz repetitive transcranial magnetic stimulation to the right parietal cortex. International Journal of Neuropsychopharmacology. 13(9), 1155-1161.

37. Nitsche M.A., Koschack J., Pohlers H., Hullemann S., Paulus W., Happe S. (2012). Effects of frontal transcranial direct current stimulation on emotional state and processing in healthy humans. Frontiers in Psychiatry. 3:58, 1-10.

38. Hung G.C. and Huang M. (2017). Transient anger attacks associated with bifrontal transcranial direct current stimulation. Brain Stimulation. 10(5), 981-982.

39. Kelley N.J., Hortensius R., Harmon-Jones E. (2013). When anger leads to rumination: induction of relative right frontal cortical activity with transcranial direct current stimulation increases anger-related rumination. Psychological Science. 24(4), 475-481.

40. Hofman, D. and Schutter, D.J. (2009). Inside the wire: aggression and functional interhemispheric connectivity in the human brain. Psychophysiology, 46(5), 1054–1058.

41. Yang CC., Völlm B., Khalifa N. (2018). The Effects of rTMS on Impulsivity in Normal Adults: a Systematic Review and Meta-Analysis. Neuropsychology Review. 28(3), 377-392.

42. Reyes-López J., Ricardo-Garcell J., Armas-Castañeda G., García-Anaya M., Arango-De Montis I., González-Olvera J.J., Pellicer F. (2018). Clinical improvement in patients with borderline personality disorder after treatment with repetitive transcranial magnetic stimulation: preliminary results. Brazilian Journal of Psychiatry. 40(1), 97-104.

43. Fischer, R., Ventura-Bort, C., Hamm, A. and Weymar, M. (2018). Transcutaneous vagus nerve stimulation (tVNS) enhances conflict-triggered adjustment of cognitive control. Cognitive, Affective, & Behavioral Neuroscience, 18(4), 680-693.

58. Rabipour S., Wu A.D., Davidson P.S.R., Iacoboni M. (2018). Expectations may influence the effects of transcranial direct current stimulation. Neuropsychologia. 119, 524-534.

66. Peña-Gómez C., Vidal-Piñeiro D., Clemente I.C., Pascual-Leone Á., Bertrés-Faz D. (2011). Down-Regulation of Negative Emotional Processing by Transcranial Direct Current Stimulation: Effects of Personality Characteristics. PLOS ONE. 6(7), 1-9.

73. Zwissler B., Sperber C., Aigeldinger S., Schindler S., Kissler J., Plewnia C. (2014). Shaping memory accuracy by left prefrontal transcranial direct current stimulation. Journal of Neuroscience. 34, 4022–4026.

74. Pirulli C., Fertonani A., Miniussi C. (2014). Is neural hyperpolarization by cathodal stimulation always detrimental at the behavioral level? Frontiers in Behavioral Neuroscience. 8, 1-10.

Reviewer 2 Report

This article is the first review that addresses this interesting topic.

The research methodology is well described and is suitable for the purpose of the work.

The paper is well organized and presents precisely the concept and measuring tools of the concept of aggression.

Before publication, the authors should correct :

  • An error on the numbering of the reference (two references with the number 1)
  • I would suggest  to replace the following reference:

  • Lam Chan.. rTMS for treatment resistant depression (2008°

 By

Gaynes BN, Lloyd SW, Lux L, Gartlehner G, Hansen RA, Brode S, Jonas DE, Swinson Evans T, Viswanathan M, Lohr KN. Repetitive transcranial magnetic stimulation for treatment-resistant depression: a systematic review and meta-analysis. J Clin Psychiatry. 2014 May;75(5):477-89; quiz 489. doi: 10.4088/JCP.13r08815. PMID: 24922485.

  •  

Author Response

Dear editor,

We are grateful for your invitation to revise and resubmit our manuscript ‘Modulating behavioural- and self-reported aggression with non-invasive brain stimulation: A literature review’, brainsci-1547015. Below, we provide a detailed point-by-point reply to all the issues that were raised by the two reviewers.

Please note that, as pointed out by reviewer #1, in the original manuscript the page numbers and line numbers did not properly show. They have now been corrected in the revised manuscript and so now the correct page numbers and line numbers are referred to when addressing the comments of the reviewers to ease verification in the revised manuscript. Please find attached the manuscript with all changes highlighted.

Reply to reviewer #2:

This article is the first review that addresses this interesting topic.

The research methodology is well described and is suitable for the purpose of the work.

The paper is well organized and presents precisely the concept and measuring tools of the concept of aggression.

Reply: We thank the reviewer for these positive words on our methodology and the organisation and presentation of our manuscript. All comments of the reviewer have been considered and have been fully incorporated in the revised manuscript.

Before publication, the authors should correct:

  • An error on the numbering of the reference (two references with the number 1)

Reply: We thank the reviewer for pointing this out. The second citation that was wrongly also noted as “citation 1” is corrected to be the second citation and all following citations have been fittingly adapted and checked.

  • I would suggest to replace the following reference:
  • Lam Chan. rTMS for treatment resistant depression (2008°)

By: Gaynes BN, Lloyd SW, Lux L, Gartlehner G, Hansen RA, Brode S, Jonas DE, Swinson Evans T, Viswanathan M, Lohr KN. Repetitive transcranial magnetic stimulation for treatment-resistant depression: a systematic review and meta-analysis. J Clin Psychiatry. 2014 May;75(5):477-89; quiz 489. doi: 10.4088/JCP.13r08815. PMID: 24922485.

Reply: We thank the reviewer for putting forward a more recent and relevant meta-analytical study on treatment-resistant depression. We agree that the reviewer’s reference fits excellently and so we have replaced our reference with the reviewer’s suggestion in line 54.

We hope that we were able to address the feedback of the reviewer in a satisfactory way. We look forward to hearing your opinion on the suitability of our revised manuscript in your special issue on Dimensions of Pathological Aggression: From Neurobiology to Therapy, for Brain Sciences.

Sincerely, also on behalf of the co-authors

Ruben Knehans

Round 2

Reviewer 1 Report

I think the manuscript greatly improved after the Authors revision. I am convinced that this final version deserves the publication in Brain Sciences.